# Protective Effects of N-Acetylcysteine on Lipopolysaccharide-Induced Respiratory Inflammation and Oxidative Stress

**DOI:** 10.3390/antiox11050879

**Published:** 2022-04-29

**Authors:** Hongbai Chen, Nana Ma, Xiaokun Song, Guozhen Wei, Hongzhu Zhang, Jing Liu, Xiangzhen Shen, Xiangkai Zhuge, Guangjun Chang

**Affiliations:** 1Ministry of Education Joint International Research, Laboratory of Animal Health and Food Safety, College of Veterinary Medicine, Nanjing Agricultural University, Weigang 1, Nanjing 210095, China; 2019807148@njau.edu.cn (H.C.); t2021153@njau.edu.cn (N.M.); 2020807166@njau.edu.cn (X.S.); 2021207061@njau.edu.cn (G.W.); 2021207039@njau.edu.cn (H.Z.); xzshen@njau.edu.cn (X.S.); 2College of Animal Husbandry and Veterinary Medicine, Jiangsu Vocational College of Agriculture and Forestry, Jurong 212400, China; 2016107100@njau.edu.cn; 3Department of Nutrition and Food Hygiene, School of Public Health, Nantong University, Nantong 226019, China; zhugexk@ntu.edu.cn

**Keywords:** N-acetylcysteine, lipopolysaccharide (LPS), acute lung injury (ALI), bovine respiratory disease (BRD), EBTr, inflammation, oxidative stress

## Abstract

As the leading cause of bovine respiratory disease (BRD), bacterial pneumonia can result in tremendous losses in the herd farming industry worldwide. N-acetylcysteine (NAC), an acetylated precursor of the amino acid L-cysteine, has been reported to have anti-inflammatory and antioxidant properties. To explore the protective effect and underlying mechanisms of NAC in ALI, we investigated its role in lipopolysaccharide (LPS)-induced bovine embryo tracheal cells (EBTr) and mouse lung injury models. We found that NAC pretreatment attenuated LPS-induced inflammation in EBTr and mouse models. Moreover, LPS suppressed the expression of oxidative-related factors in EBTr and promoted gene expression and the secretion of inflammatory cytokines. Conversely, the pretreatment of NAC alleviated the secretion of inflammatory cytokines and decreased their mRNA levels, maintaining stable levels of antioxidative gene expression. In vivo, NAC helped LPS-induced inflammatory responses and lung injury in ALI mice. The relative protein concentration, total cells, and percentage of neutrophils in BALF; the level of secretion of IL-6, IL-8, TNF-α, and IL-1β; MPO activity; lung injury score; and the expression level of inflammatory-related genes were decreased significantly in the NAC group compared with the LPS group. NAC also ameliorated LPS-induced mRNA level changes in antioxidative genes. In conclusion, our findings suggest that NAC affects the inflammatory and oxidative response, alleviating LPS-induced EBTr inflammation and mouse lung injury, which offers a natural therapeutic strategy for BRD.

## 1. Introduction

Due to high morbidity and economic ramifications, the bovine respiratory disease (BRD) complex is one of the biggest health obstacles faced by the cattle industry [1]. It is a complex of diseases characterized by many types of infection, each having its own causes, clinical signs, and economic implications. Bacterial infection is a prevalent cause of BRD, and many pathogens are Gram-negative bacteria, including *Mannheimia haemolytica, Pasteurella multocida,* and *Histophilus somni* [2,3]. Antibiotics are preferred to control BRD in practice [4], but the adverse effects caused by the overuse of antibiotics cannot be ignored in the future, and there is an urgent need to apply effective and environment-friendly antibiotic alternatives to keep animals healthy and prevent infection in the cattle industry.

Lipopolysaccharide (LPS), a vital component of the outer cell wall of Gram-negative bacteria, is thought to be one of the main causes of inflammation [5,6,7]. Circulating LPS as a pathogen-associated molecular pattern (PAMP) can stimulate the innate immune system and mediate a local or systemic inflammatory response. LPS can also stimulate non-immune cells and initiate the inflammatory process. After reacting with the immune system, inflammatory cytokines such as TNF-α, IL-6, IL-1β, and IL-8 can be upregulated. IL-8 plays an essential role in initiating and amplifying the inflammatory response [8,9].

A large number of studies have confirmed that the intratracheal administration of LPS can effectively induce acute lung injury [10,11]. Most animal models of ALI are characterized by a prominent neutrophilic infiltration into alveolar space and interstitial septa thickness [10,11,12]; these pathological features are similar to those of cattle that suffer from bacterial pneumonia. 

Oxidative stress is defined as an imbalance between free radical production, reactive metabolites, and their elimination by protective mechanisms [13]. This imbalance leads to the damage of important tissues and organs, with a potential impact on the whole organism [14]. A variety of transcription factors can be activated by oxidative stress, and some genes involved in inflammatory pathways can be upregulated and trigger immunoreaction. An appropriate inflammatory response is conducive for animals to overcome infection, but a systemic inflammatory reaction to endotoxins can lead immune cells to produce excessive amounts of cytokines or reactive oxygen species (ROS), resulting in systemic tissue injury. Neutrophils play an essential role in bacteriolysis. After being recruited in inflammation, vast quantities of reactive oxygen species, such as O^2•–^, are produced by neutrophils through their oxidant-generating systems, such as the phagocyte NADPH oxidase or nitric oxide synthase (NOS) [15]. ROS production by phagocytes has well-known antimicrobial activities during phagocytosis and was first referred to as respiratory burst. It is characterized by a rapid and cyanide-insensitive increase in oxygen uptake, increase in glucose consumption, and the immediate release of ROS [16]. Hydroxyl radical and hypochlorous acid are generated during respiratory burst, which both contribute to the death and destruction of bacteria within the phagosome. However, the excessive production of ROS promotes endothelial dysfunction by the oxidation of crucial cellular signaling proteins such as tyrosine phosphatases, which will eventually result in a disorder in the internal environment and cause tissue injury. 

To prevent ROS-induced injury, mammalian cells have evolved an array of antioxidant defense systems. The antioxidant enzymes superoxide dismutase (SOD), catalase (CAT), and glutathione peroxidase (GPX) are classified as ROS scavengers [17]. Oxidative stress occurs when enhanced oxidant production overwhelms the capacity of antioxidant enzymes. Vertebrate cells have different isoforms of SOD: SOD1, known as CuZn SOD, a 32kD homodimer expressed in the cytosol and nucleus; and SOD2, or MnSOD, which exists in human mitochondria, and can form different chemical bonds depending on the Mn oxidation state [17,18]. All SODs are highly expressed in lung tissue, vessels, and airways [19]. Compared with their relative distribution, the activities of CuZnSOD and MnSOD are lower in the lung [20]. CuZnSOD and MnSOD have higher activity levels in the liver, kidney, heart, and brain. Previous studies indicated that the knockout MnSOD-deficient mice die in the neonatal stage from dilated cardiomyopathy and impaired neural development [21,22].

Nrf2, a basic leucine zipper protein, plays an important role in regulating the induction of several antioxidant and cytoprotective genes. In response to oxidative stress, Nrf2 dissociates from Keap1 and translocates into the nucleus, activating the translation and expression of antioxidant-related factors. Nrf2 deficiency enhances the susceptibility to experimental acute lung injury and impairs the resolution of lung inflammation in mice [23]. 

N-acetylcysteine (also known as NAC) has been used in clinical practice for several decades. As the acetylated precursor of the amino acid L-cysteine, NAC has a relatively low toxicity and is related to mild side effects, such as nausea, vomiting, rhinorrhea, pruritus, and tachycardia [24]. It has been reported that, after NAC treatment, a significant improvement in the expression of the Nrf2 gene and antioxidant enzyme was observed compared with a control group [25]. This phenomenon showed that NAC supplementation could protect against oxidative stress by enhancing Nrf2 expression. In addition, the continuous administration of high-dose NAC ameliorated hemorrhagic shock induced by acute lung and kidney injury, decreasing histopathologic scores, malondialdehyde (MDA) levels, and the levels of other inflammatory factors in rats [26]. A previous study reported that the intravenous injection of liposomal-formulated NAC successfully improved prophylactic effectiveness against LPS-induced lung injuries [27]. Most past studies gave NAC intravenously [26,27,28], while, in daily practice, it is still unclear whether NAC could be given as a feed supplement in the dairy industry to alleviate the inflammatory response or tissue damage caused by lipopolysaccharide.

By reviewing the literature described above, it can be seen that NAC has a regulatory effect on LPS-induced inflammation. Currently, there is little research about whether NAC can protect ruminants from LPS-induced pneumonia. In this study, we established an LPS-induced inflammatory model in embryonic bovine tracheal cells (EBTr) and an acute lung injury (ALI) model in ICR (Institute of Cancer Research) mice to explore the potential mechanism of the NAC in moderating the inflammatory response and oxidative stress in LPS-induced injury models.

## 2. Materials and Methods

### 2.1. Ethics Statement

The experiment design was approved by the Institutional Animal Care and Use Committee of Nanjing Agricultural University and all procedures in this experiment strictly followed the “Guidelines for Experimental Animals” of the Ministry of Science and Technology (2006, Beijing, China). 

### 2.2. Materials

Dexamethasone (D4902) and LPS (L2880, from E. coli 055:B5) were purchased from Sigma-Aldrich (St. Louis, MO, USA). N-acetylcysteine (A9165-25G) was obtained from Thermo Fisher Scientific Inc. (Waltham, MA, USA). Chloral hydrate (R24095) was acquired from Shanghai yuanye Bio-Technology Co., Ltd. (Shanghai, China). Wright–Giemsa Stain solution was purchased from Beijing Solarbio Science & Technology Co., Ltd. (Beijing, China). The other reagents used were of analytical grade.

### 2.3. Establishment of Lung Injury Model

Sixty 7- to 9-week-old female ICR mice weighing 25–30 g (Qinglongshan Animal Cultivation Farm, Nanjing, Jiangsu) were quarantined for 7 days. After adaption to the environment, mice were randomly assigned to the following five groups (n = 12 per group): control (PBS), NAC, LPS, LPS+NAC, and DEX (dexamethasone, as a positive drug, 5 mg/kg dissolved in PBS) + LPS (Figure 1). Mice were intragastrically administered NAC (600mg/kg) or PBS as a control for three consecutive days. At the last administration, all mice received the last NAC or PBS gavage, and the LPS+DEX group received DEX (5mg/kg) intraperitoneally. After 2 hours, mice were anesthetized with 10% chloral hydrate (0.06 mL/10 g; IP), and LPS (5mg/kg, dissolved in 50μL PBS) was administered intratracheally. The mice in the control and NAC groups received the same volume of PBS. After 12 h of LPS administration, the animals were euthanized by overdose IP injection of 10% chloral hydrate. Subsequently, lung tissue samples and the bronchoalveolar lavage fluid (BALF) were collected for research use.

### 2.4. Experimental Design In Vitro

First, EBTr cells were treated with different concentrations of LPS and NAC solutions to obtain a suitable drug range for further experiments. Then, LPS was administrated to EBTr cells for different lengths of time to explore its effect on inflammation and oxidative stress. The expression of IL-6, IL-8, and TNF-α was measured for inflammation, and glutamate–cysteine ligase catalytic subunit (GCLC), NAD(P)H quinone dehydrogenase 1 (NQO1), heme oxygenase-1 (HO-1), and nuclear factor-erythroid factor 2-related factor 2 (Nrf2) for oxidative stress. To study the protective properties of NAC, 1 mmol/L NAC was chosen to treat cells with for 6 h, and, after PBS washing, LPS was administrated for 6 h. The anti-inflammatory and oxidative stress properties of NAC were explored through qPCR and cytokine measurements.

### 2.5. Cell Culture and CCK8 Assay

EBTr cells were cultured in DMEM basic medium supplemented with 15% fetal bovine serum and 1% penicillin-streptomycin (J180014, HyClone, Logan, UT, USA) in 37℃ and 5% CO_2_. In all experiments, cells were allowed to acclimate for 24 h before any treatments

Cell viability was measured by a CCK8 assay by following the manufacturer’s instructions. In a 96-well plate, 5 × 10^3^ cells were inoculated for 24 h, after which, different concentrations of LPS (1, 2, 4, 8, or 16 μg/mL) and NAC (0, 0.25, 0.5, 1, 5, 10 mmol/L) were added to the culture medium for 24 h. Then, the CCK-8 Cell Proliferation and Cytotoxicity Assay Kit (Solarbio Life Sciences, Beijing, China) was used to access the survival rate of EBTr cells according to the manufacturer’s instructions.

### 2.6. Cell Counting and Protein Concentration Assay in Bronchoalveolar Lavage Fluid (BALF)

After LPS challenge for 12 h, six mice in each group (n = 12) were randomly selected for the collection of BALF. A total of 0.8 mL of 4 °C PBS was gently injected into the lung through 1 mL syringe, and the solution was aspirated slowly to ensure a volume of lavage fluid of at least 0.6 mL. BALF samples were centrifuged to pellet the cells, lysed by ACK Lysis Buffer (Beijing Solarbio Science & Technology Co., Ltd., Beijing, China) for 10 min, washed twice with ice-cold PBS and collected, and then again centrifuged for another 5 min at 4 °C. Subsequently, the sedimented cells were resuspended in PBS to obtain the total counts of cells, neutrophils, and macrophages, which were obtained using a hemocytometer, and the Wright–Giemsa staining method was used for cytosine staining. In addition, the supernatants were collected and their protein concentrations were directly measured using a BCA protein assay kit (Thermo Fisher Scientific, Waltham, MA, USA).

### 2.7. Histopathological Evaluation

A histopathological examination was performed on the mice that were not subjected to BALF collection. The left lungs were isolated and immersed in 4% paraformaldehyde. After 24 h of fixation, the samples were dehydrated in a series of graded ethanol, embedded in paraffin wax, cut into 5 μm-thick sections, and stained with hematoxylin and eosin (H & E) for pathological analysis. The pathological changes were evaluated and scored according to the protocol described previously [29].

### 2.8. Measurement of Myeloperoxidase (MPO), MDA, CAT and SOD Levels in Lung Tissues

All mice were sacrificed 12 h after LPS intratracheal injection, and the right lungs were excised and homogenized for analysis of MPO, MDA, SOD, and CAT content (Cat. No. A044-1-1 MPO, A003-1-2 MDA, A001-3 SOD, A007-1-1 CAT, Jiancheng Bioengineering institute, Nanjing, China). Further description of these methods is provided in Appendix A. As a part of the host defense system of polymorphonuclear leukocytes, studying the activity of MPO can be used to examine neutrophil accumulation in lung tissue. MDA content was assessed to measure the level of lipid peroxidation. Furthermore, to measure the antioxidative enzyme activities in the lung tissue, SOD and CAT levels were detected following the respective manufacturer’s instructions.

### 2.9. ELISA for Cytokines

The BALF was obtained from mouse models and centrifuged, and supernatants were collected for measurement of the TNF-α, IL-6, and IL-1β secretion using an enzyme-linked immunosorbent assay (ELISA) kit, respectively, and following the manufacturer’s instructions (CK-EN20174, CK-EN 20188, and CK-EN20852, Nanjing RuiXin Biological Technology Co., Ltd., Nanjing, China).

Supernatants of cell culture medium were also collected for IL-6 and IL-8 contents analysis. Enzyme-linked immunosorbent assay (ELISA) kits were used (Bovine IL-6, CK-EN77030; Bovine IL-8, CK-EN77031, Nanjing RuiXin Biological Technology Co., Ltd., Nanjing, China) according to the manufacturer’s instructions.

### 2.10. RNA Extraction and Quantitative Real-Time Polymerase Chain Reaction (qRT-PCR)

Total RNA was obtained using TRIzol reagent (cat. 9108, Takara, Dalian, China). Agarose gel (1%) electrophoresis and a NanoDrop ND-1000 Spectrophotometer (Thermo Scientific, Waltham, MA, USA) were used to assess RNA quality. Reverse transcription (RT) was performed using 250 ng/μL RNA to synthesize cDNA with HiScript^®^III RT SuperMix for qPCR (+gDNA wiper) kit (R323, Vazyme Biotech Co., Ltd., Nanjing, China) according to the manufacturer’s instructions. All primers used in this experiment are listed in Table 1 and Table 2. GAPDH was chosen as the house-keeping gene to normalize the data of each target gene. cDNA samples were amplified using Cham Q Universal SYBR qPCR Master Mix (Q711, Vazyme Biotech Co., Ltd., Nanjing, China) and an ABI 7300 Fast real-time PCR system (Applied Biosystem, USA). Briefly, the reaction mixture consisted of 1 μL of cDNA and 0.2 μM primers, 5 μL of SYBR green, and 3.8 mL of ddH2O in a final volume of 10 μL. The relative expression of each target gene was analyzed using the 2-ΔΔCt method as described previously [30]. The results are presented as fold changes relative to control.

### 2.11. Statistical Analysis

All data referenced above were expressed as the means ± SEM and analyzed using SPSS23.0 (IBM). Each experiment was carried out in at least triplicate. The data comparison between experimental group was conducted applying one-way ANOVA. Multiple comparisons were made using the LSD method. The normality distribution of the variables was checked by the Shapiro–Wilk test of SPSS before difference comparison. All probability values were 2-sided. Statistical significance for all studies was accepted as *p* < 0.05.

## 3. Results

### 3.1. Effect of Different Concentrations of LPS and NAC on the EBTr Viability

The viability of EBTr decreased when treated with 16 μg/mL LPS (*p* < 0.01), but not at concentrations of 0, 1, 2, 4, and 8 μg/mL (Figure 2A), indicating that 1 to 8 μg/mL of LPS had no effect on cell viability and could be used in subsequent experiments. The cell viability decreased at a concentration of 5 mM NAC (*p* < 0.01) (Figure 2B), whereas no difference was observed at 0, 0.25, 0.5, and 1 mM compared with that of the control group. These results show that the concentrations of LPS and NAC used in the following experiments did not affect the EBTr viability.

### 3.2. Inflammation and Oxidative Stress during LPS Administration at Different Times and the Effect of NAC Supplementation

The secretion of IL-6 (*p* < 0.05) and IL-8 (*p* < 0.01) in the culture medium reached a peak at 6 h of LPS treatment (Figure 3). While the expression of inflammatory genes followed the same trend, IL-6 (*p* < 0.01) and IL-8 (*p* < 0.01) peaked at 6 h. We also observed that cells treated with 6 h LPS had significant changes in the expression of HO-1 (*p* < 0.01), Nrf2 (*p* < 0.01), and NQO1 (*p* < 0.05) mRNA levels. These results indicate that a 6 h incubation time can be selected to induce oxidative stress and inflammation. A total of 1 mM NAC could reverse the change in HO-1 (*p* < 0.01), Nrf2 (*p* < 0.01), and NQO-1 (*p* < 0.05) mRNA levels. Furthermore, we observed that the expression and secretion of IL-6 (*p* < 0.01) and IL-8 (*p* < 0.01) induced by 6 h LPS exposure could be downregulated after 1 mM NAC pretreatment.

### 3.3. NAC Pretreatment Moderated the LPS-Induced Histopathological Changes in ALI Mice

The histological changes in lung tissues were assessed by light microscopy, and the lung injury scores were determined. As illustrated in Figure 4, no obvious abnormalities were observed in the control and NAC groups. In the LPS group, the alveolar space collapsed, with intramural red blood cells and clusters of intramural neutrophils accompanied by a dramatically increased lung injury score (*p* < 0.01). In contrast, pathological improvement was observed in the NAC or DEX pretreatment groups; this change is consistent with a decreased lung injury score (*p* < 0.01).

### 3.4. NAC Treatment Ameliorated Cytokine Secretion, Protein Leakage, Total Cell Numbers, and Neutrophils in BALF and MPO Levels in LPS-Induced ALI Mice

Mice were euthanized 12 h after LPS injection to obtain their lung tissues and BALF. We measured the relative protein concentration, inflammatory factor levels, and total cells in BALF. NAC and DEX administration successfully inhibited the secretion levels of IL-6, TNF-α, and IL-1β induced by LPS stimulation (Figure 5E). Mice exposed to LPS have a greater level of cell infiltration (Figure 5A), especially neutrophils (Figure 5B), whereas pretreatment with NAC and DEX effectively inhibited the LPS-induced increase in these cells (*p* < 0.01). The severity of edema formation was educated by protein leakage in BALF. As illustrated in Figure 5C, LPS-stimulated protein concentrations were significantly higher than those of the normal and NAC-treated group. The pretreatment of DEX remarkably inhibited the protein leakage. MPO analysis can examine neutrophil activities in lung tissue. MPO levels were significantly decreased in the LPS+NAC and LPS+DEX group compared with the LPS-alone group (Figure 5D). 

### 3.5. Influence of NAC on Antioxidant Enzyme Activity and Lipid Peroxidation in Mice

The values of CAT, SOD, and MDA in the lungs are shown in Figure 6. Increased MDA levels represent the evaluated oxidative injury in the lungs. The administration of NAC significantly decreased lung tissue MDA levels compared with the LPS group. Additionally, the results show significant increases in SOD and CAT activities in the NAC group, indicating that NAC and DEX pretreatment could inhibit LPS-triggered suppression antioxidant enzymes.

### 3.6. NAC Reverses Inflammatory and Oxidative-Stress Related Genes Changes in ALI Mice

The gene expression related to inflammation is shown in Figure 7. LPS stimulation induced increases in TNF-α, IL-1β, IL-6, and IL-8 gene expression levels compared with the control group, whereas NAC pretreatment successfully suppressed this trend. NAC administration can significantly increase the mRNA abundance of CAT, Nrf2, and SOD-1 compared with the control group (Figure 8). On the other hand, LPS stimulation can significantly reduce the mRNA expression of GCLC, Nrf2, CAT, and HO-1 in mouse lung tissues (Figure 8A–C,E).

## 4. Discussion

LPS, as a component of the cell wall of Gram-negative bacteria, is considered to be one of the most important promoters of pneumonia and systemic inflammation [31]. Several studies have shown that the cascading amplification of the inflammatory response, or a “systemic cytokine storm”, is one of the pathophysiological mechanisms of acute lung injury [32,33]. Due to the fact that LPS is an initiating factor of the inflammatory response, we selected it in mice to simulate acute lung injury. The main characteristics of LPS-induced ALI in mice include an increased pulmonary microvascular permeability, diffuse pulmonary interstitial and alveolar edema, and massive inflammatory cell infiltration [34]. Card, J W et al. reported that male mice were more susceptible to lipopolysaccharide-induced lung injury [35]. For this reason, to eliminate experimental variation as a result of gender differences, only female mice were used in this study. In our experiments, after LPS administration, the number of Toll cells and neutrophils was remarkably increased in BALF. Histopathological analysis showed that LPS injection caused severe alveolar septal thickening, collapse of the alveolar space, and lung hemorrhage. We believe that LPS-induced ALI models were successfully established. 

Previous reports showed that NAC treatment reduced the severity of H9N2 swine influenza virus-induced ALI and may have beneficial effects in the prevention of H9N2 swine influenza virus infection [36]. The intraperitoneal administration of NAC successfully corrected unbalanced cytokines such as Th1/Th2/Th17 and downregulated Galectin-9/Tim-3 expression in ALI mouse models [26,37]. In our experiments with the pretreatment of NAC, the numbers of Toll cells and neutrophils were significantly decreased, followed by alleviated pathological changes and lower grades of lung injury score. The dosage form of drugs is also important. The encapsulation of NAC increased its protective effects against intracellular damage. A previous study reported that liposomal-N-acetylcysteine is more effective than NAC against lung injury [27]. The treatment effect was attributed to the ability of liposomes to deliver higher levels of the antioxidant to the lung. This provides insight toward the future improvement of NAC efficacy against BRD.

Prominent neutrophilic infiltration into the alveolar space and interstitial septa is the main characteristic of ALI and subsequent tissue injury [38]. To assess the levels of neutrophils in the entire volume, bronchoalveolar lavage was conducted, and BALF was collected to acquire total and differential cell counts. While the assessment of the total neutrophil numbers does not give information about the activation state of counted cells, measurements of myeloperoxidase (MPO) activity play a vital role in revealing the functional state of this cell population [12]. MPO is present in all cells of the myeloid lineage, but it is most abundant in the azurophilic granules of neutrophils [39]. In the present study, total cells and neutrophils were remarkably increased in BALF, accompanied by dramatically elevated MPO activity in the LPS group. This result is in line with the histopathological analysis. However, LPS treatment induced severe changes in cell numbers, and MPO levels were obviously weakened by pretreatment with NAC or DEX. The results suggest that the addition of NAC alleviated lung injury by reducing the infiltration of neutrophils and inhibiting MPO activities.

During the inflammatory response, cytokines released from macrophages or polymorphonuclear leukocytes, including TNF-α, IL-1β, IL-6, and IL-8, are inextricably linked to inflammation and neutrophilic infiltration. A crucial role of NAC in regulating the secretion of inflammatory cytokines has been reported [40]. The intraperitoneal administration of NAC can inhibit the increase in the BALF concentrations of TNF-α, IL-6, IL-1β, and IL-8 in H9N2 swine influenza virus-induced ALI mouse models [36]. In the current study, the secretion and gene expression of TNF-a, IL-6, and IL-1β were determined. Although LPS injection remarkably upregulated the mRNA and protein levels of TNF-a, IL-6, and IL-1β, NAC pretreatment successfully reversed this trend and maintained homeostasis. 

Some reports observed that endotoxin exposure resulted in the formation of superoxide radicals, followed by the formation of hydrogen peroxide, hydroxyl radicals, and other forms of ROS [41,42]. In contrast, there are antioxidant systems that exist in mammals to handle ROS overproduction. Catalases are well-studied enzymes that play critical roles in protecting cells against the toxic effects of hydrogen peroxide, being able to utilize their heme group to reduce/oxidize H2O2, relieving oxidative stress [43]. Superoxide generated in vivo can be directly scavenged by superoxide dismutase (SODs), including copper–zinc superoxide dismutase (Cu/ZnSOD), manganese superoxide dismutase, and extracellular superoxide dismutase [44]. In our study, the LPS group suffered low levels of SOD and CAT activities, which corresponded to lower expressions of antioxidant-related mRNAs, including those for CAT, HO-1, Nrf2, and GCLC. Supplementation with NAC or DEX promotes antioxidant activities, which was reflected by increased CAT and SOD contents and a decreased MDA level. MDA is a highly toxic by-product generated by lipid peroxidation, and its toxicity depends on its rapid reaction with proteins and DNA [45]. If accumulated, these lipid peroxy radicals act on nearby fatty acids in the plasma membranes of the cells and induce radical formation by a positive feedback loop [46]. To study the endotoxin-induced injury model in vitro, we used LPS to induce inflammatory injury in EBTr cells, and investigated the role of NAC in LPS-induced EBTr cells. Our results reveal the optimal LPS incubation time (6 h) for inducing inflammation. LPS effectively upregulated the expression of protein and the mRNA levels of inflammatory-related factors, which also resulted in changes in antioxidant-related genes. These findings demonstrate that LPS effectively induced inflammatory injury in EBTr cells. NAC pretreatment ameliorated the increase in IL-6 and IL-8 cytokines and mRNA abundance. Nf-E2-related factor-2 (Nrf2) is a basic leucine zipper transcription factor that binds to and activates the antioxidant response element (ARE) in the promoters of many antioxidant and detoxification genes [47]. By upregulating the mRNA abundance of antioxidative factors such as Nrf2 and NQO-1, NAC exerted its antioxidative ability in embryonic bovine tracheal cells. 

We cannot ignore the fact that some results revealed in EBTr cells were not reproduced in the LPS-ALI model. Consistent with other studies, LPS administration induced HO-1 upregulation in EBTr cells [48]. ALI mice, as a more complex system, do not follow this trend. We hypothesize that, at an early stage of infection, HO-1 inhibition could help tissue to recruit more inflammatory cells in the injury site. Theoretically, NAC or DEX could exert their antioxidant effects through activating the Keap1/Nrf2/HO-1 axis, a pathway that has been established by many studies [49,50]. While, in our experiment, even a strong anti-inflammatory and antioxidant effect was proven by various results (such as histopathological changes, MPO activity, MDA content, antioxidant enzyme activities, and the analysis of BALF), factors such as CAT, Nrf2, SOD-1, and GCLC did not differ between the LPS and drug pretreatment groups (LPS+NAC and LPS+DEX). This may suggest that the antioxidant effects of NAC and DEX do not depend solely on the Nrf2 pathway. The mechanisms and the physiological significance underlying these changes need to be further investigated.

## 5. Conclusions

In conclusion, the LPS challenge induced oxidative stress and inflammatory responses in both EBTr cells and mouse models, indicating the disruption of cell homeostasis, thus resulting in acute lung injury, whereas NAC alleviated inflammatory states and oxidative stress. The molecular mechanisms underlying the anti-inflammatory effects of NAC include (1) the inhibition of the generation of proinflammatory cytokines, such as TNFα, IL-1β, and IL-6, and (2) the reduction in ROS production by the attenuation of the activity of antioxidant enzymes (AOEs). We aim to offer a natural therapeutic strategy for BRD caused by LPS instead of antibiotics.

## Figures and Tables

**Figure 1 antioxidants-11-00879-f001:**
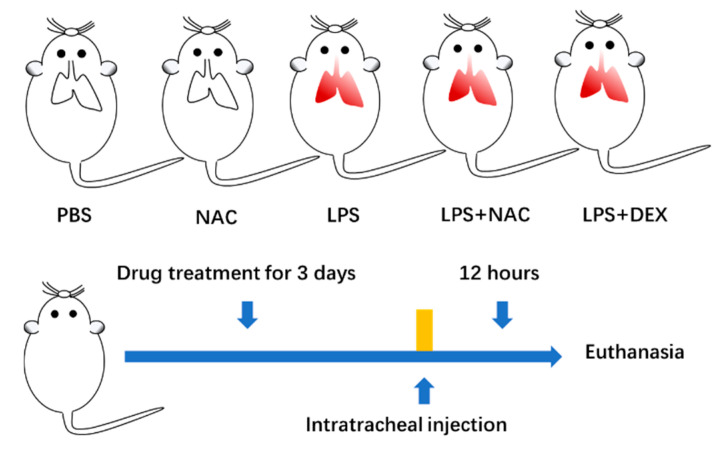
Mice were divided into five groups. After drug treatment for 3 consecutive days, the intratracheal injection was conducted, and they were euthanized 12 h later to collect samples.

**Figure 2 antioxidants-11-00879-f002:**
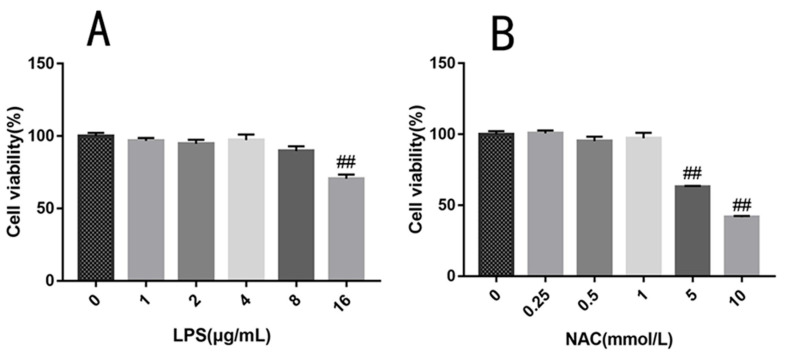
Evaluation of cell viability following different concentrations of LPS and NAC treatment. Cells were exposed to different concentrations of (**A**) LPS (1, 2, 4, 8, or 16 μg/mL) and (**B**) NAC (0, 0.25, 0.5, 1, 5, 10 mmol/L) for 24 h. All results are expressed as the means ± SEM. ## indicates *p* < 0.01.

**Figure 3 antioxidants-11-00879-f003:**
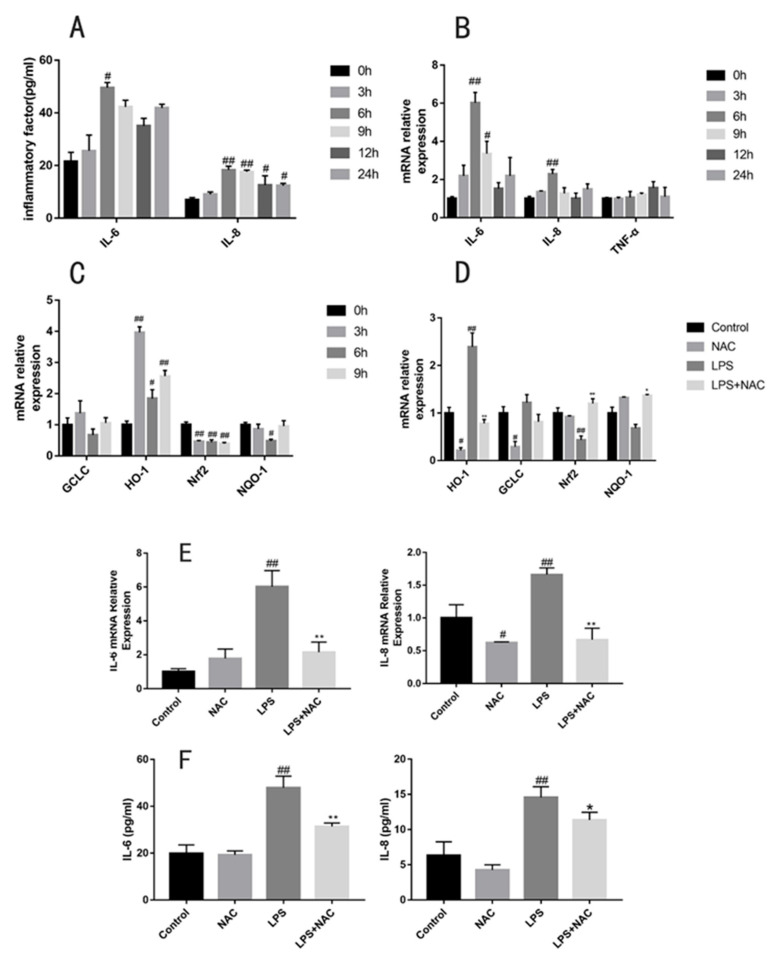
Gene expression and secretion of cytokines in bovine embryo tracheal cells (EBTr) exposed to LPS (4 μg/mL) at different times. (**A**) Secretion of IL-6 and IL-8 and (**B**) gene expression of IL-6, IL-8, and TNF-α detected after treatment with 4 μg/mL LPS for 0, 3, 6, 9, 12, 24 h. (**C**) Relative mRNA levels of antioxidants (GCLC, Nrf2, HO-1, NQO-1) were measured at 0, 3, 6, and 9 h of 4 μg/mL LPS exposure. (**D**) Using q-PCR to detect antioxidant gene (GCLC, Nrf2, HO-1, NQO-1) expression changes after 1 mM NAC administration. NAC pretreatment significantly decreased (**E**) gene expression levels and (**F**) the secretion of IL-6 and IL-8. All results are expressed as the means ± SEM. ## indicates *p* < 0.01, # indicates *p* < 0.05 compared with the control group; ** indicates *p* < 0.01, * indicates *p* < 0.05 compared with the LPS group.

**Figure 4 antioxidants-11-00879-f004:**
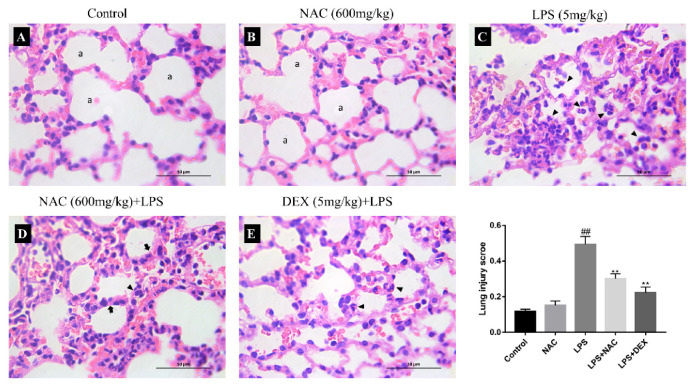
Effect of NAC and DEX on histological changes in LPS-induced ALI mouse model. (**A**–**E**) indicate the lung structure of control, NAC, LPS, NAC+LPS, and DEX+LPS groups, respectively. No obvious abnormalities were observed in (**A**,**B**) (a, alveolar space). As illustrated in (**C**), the alveolar space is collapsed with intramural accumulation of red blood cells, and there are clusters of intramural neutrophils (arrow head). (**D**) shows thickened alveolar walls (arrows) with a small number of red blood cells in the lumen and neutrophils (arrow head) in the alveolar septum. In (**E**), most of the neutrophils are located in the alveolar septum, with only a few in the alveolar space. The sections were stained with hematoxylin and eosin staining (630× magnification). The black bar = 50 μm at 630× magnification. NAC, N-acetylcysteine; LPS, lipopolysaccharide; NAC+LPS, LPS with N-acetylcysteine pretreatment; DEX+LPS, LPS with dexamethasone. All results are expressed as the means ± SEM (n = 6 in each group). ## indicates *p* < 0.01 compared with the control group; ** indicates *p* < 0.01 compared with the LPS group.

**Figure 5 antioxidants-11-00879-f005:**
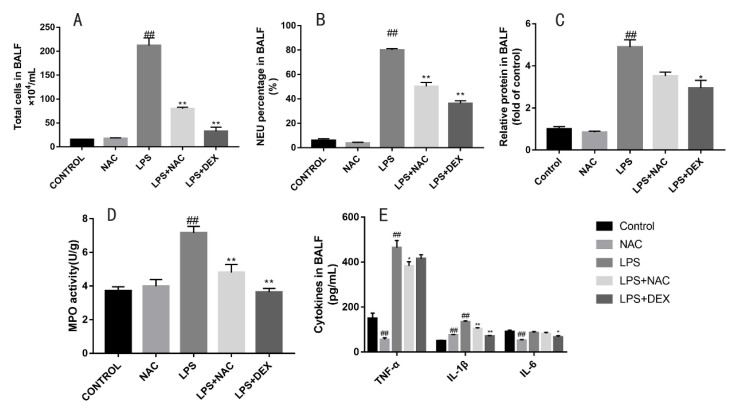
Protective effects of NAC treatment against LPS-induced ALI mice. BALF was collected 12 h after LPS challenge to measure (**A**) total cells, (**B**) neutrophil percentage, (**C**) amount of protein, and (**E**) level of cytokines secretion. MPO activity in lung tissues (**D**) was measured at 12 h after intratracheal injection. All results are expressed as the means ± SEM (n = 6 in each group). ## indicates *p* < 0.01 compared with the control group, and ** indicates *p* < 0.01, * indicates *p* < 0.05 compared with the LPS group.

**Figure 6 antioxidants-11-00879-f006:**
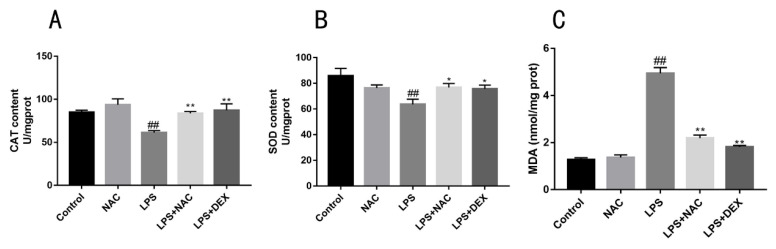
Effects of NAC treatment on LPS-triggered oxidative stress in ALI mice. Effects of NAC on levels of CAT (**A**), SOD (**B**), and MDA (**C**) from lung homogenates are shown. All results are expressed as the means ± SEM (*n* = 6 in each group). ## indicates *p* < 0.01 compared with the control group, and ** indicates *p* < 0.01, * indicates *p* < 0.05 compared with the LPS group.

**Figure 7 antioxidants-11-00879-f007:**
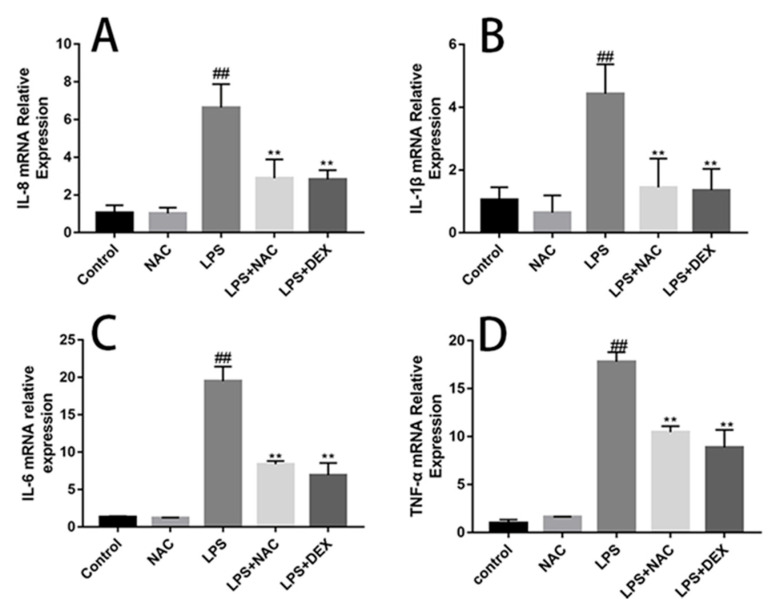
Effect of pre-treatment of NAC on the expression of inflammatory factors in lung tissue. The gene expression levels of IL-8 (**A**), IL-1β (**B**), IL-6 (**C**), and TNF-α (**D**), as shown. GAPDH was used to normalize the gene expression data. TNF-α, IL-1β, IL-8, and IL-6 expression levels were remarkably increased in the LPS group compared with those in control group. All results are expressed as the means ± SEM (n = 6 in each group). ## indicates *p* < 0.01 compared with the control group, and ** indicates *p* < 0.01 compared with the LPS group.

**Figure 8 antioxidants-11-00879-f008:**
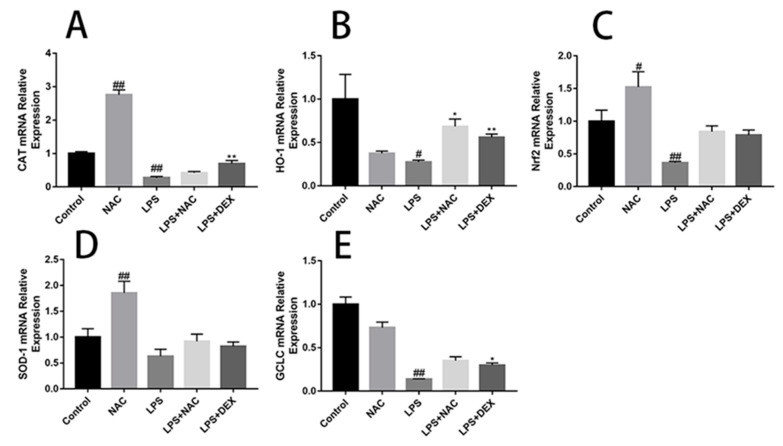
Effect of pre-treatment of NAC on the expression of factors related to oxidative stress in lung tissue. The gene expression levels of CAT (**A**), HO-1 (**B**), Nrf2 (**C**), SOD-1 (**D**), and GCLC (**E**) are shown in figures. GAPDH was used to normalize the gene expression data. The NAC group shows a significant increase in CAT, Nrf2 and SOD-1 relative mRNA expression levels compared with control group. LPS injection decreased the expression of CAT, HO-1, Nrf2, and GCLC genes. After NAC pretreatment, HO-1 relative mRNA levels increased compared with the LPS group. All results are expressed as the means ± SEM (n = 6 in each group). ## indicates *p* < 0.01, # indicates *p* < 0.05 compared with the control group, and ** indicates *p* < 0.01, * indicates *p* < 0.05 compared with the LPS group.

**Table 1 antioxidants-11-00879-t001:** PCR Primers Used for the Quantification of the Candidate Genes in mice.

Target Gene		Primer Sequence	Accession	Product (bp)
HO-1	Forward	TGAAGGAGGCCACCAAGGAGG	NM_010442.2	375
	Reverse	AGAGGTCACCCAGGTAGCGGG		
GCLC	Forward	CACTGCCAGAACACAGACCC	XM_006510812.2	238
	Reverse	ATGGTCTGCTGAGAAGCCT		
SOD-1	Forward	GTTCCACGTCCATCAG	NM_011434.2	144
	Reverse	TCCTTTCCAGCAGTCA		
Nrf2	Forward	CTGGCTGATACTACCGCTGTT	NM_010902.5	164
	Reverse	GTGGAGAGGATGCTGCTGAA		
CAT	Forward	CAGGAAGGCTTGCTCAGGAA	NM_009804.2	81
	Reverse	AGGACGGGTAATTGCCATTG		
IL-6	Forward	AGGACTCTGGCTTTGTCTTTC	NM_001314054.1	216
	Reverse	CAATGGCAATTCTGATTGTATG		
IL1-β	Forward	TCATCTCGGAGCCTGTAGTGC	XM_006498795.5	292
	Reverse	GCTGCTTCCAAACCTTTGACC		
TNF-α	Forward	GCTCTGTGAAGGGAATGGGTGT	NM_001278601.1	276
	Reverse	CCAGGTCACTGTCCCAGCATCT		
IL-8	Forward	CGGCAATGAAGCTTCTGTAT	NM_011339.2	224
	Reverse	CCTTGAAACTCTTTGCCTCA		
GAPDH	Forward	ACCACAGTCCATGCCATCAC	XM_036165840.1	452
	Reverse	TCCACCACCCTGTTGCTGTA		

**Table 2 antioxidants-11-00879-t002:** PCR Primers Used for the Quantification of the Candidate Genes in EBTr.

Target Gene		Primer Sequence	Accession	Product (bp)
HO-1	Forward	TGAAGGAGGCCACCAAGGAGG	NM_001014912	375
	Reverse	AGAGGTCACCCAGGTAGCGGG		
GCLC	Forward	ATTGGGTGGAGAGTGGAA	XM_024983548.1	133
	Reverse	ACAGCGGGATGAGAAAGT		
NQO-1	Forward	CAACAGACCAGCCAATCA	NM_001034535.1	144
	Reverse	ACCTCCCATCCTTTCCTC		
Nrf2	Forward	CTGGCTGATACTACCGCTGTT	XM_005202312.4	164
	Reverse	GTGGAGAGGATGCTGCTGAA		
CAT	Forward	TCACTCAGGTGCGGACTTTC	NM_001035386.2	163
	Reverse	TGGATGCGGGAGCCATATTC		
IL-6	Forward	GGAGGAAAAGGACGGATGCT	NM_173923.2	227
	Reverse	GGTCAGTGTTTGTGGCTGGA		
TNF-ɑ	Forward	CTTCTGCCTGCTGCACTTCG	XM_005223596.4	153
	Reverse	GAGTTGATGTCGGCTACAACG		
IL-8	Forward	CCTCTTGTTCAATATGACTTCCA	NM_173925.2	170
	Reverse	GGСССАСТСТCAТАACTCTC		
GAPDH	Forward	GGGTCATCATCTCTGCACCT	NM_001034034	177
	Reverse	GGTCATAAGТCCCTCCACGA		

## Data Availability

The data is contained with the article and Appendix A.

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
