# Peer review of "Protective Effects of N-Acetylcysteine on Lipopolysaccharide-Induced Respiratory Inflammation and Oxidative Stress"

_antioxidants, 2022, doi:10.3390/antiox11050879_

Round 1
Reviewer 1 Report
The study by Chen et al examined the protective effects of NAC in an LPS model of ALI, as a model of bovine bacterial pneumonia.
The protective effect of NAC in experimental ALI in rodents has been demonstrated in several studies. Bovine tracheal cells are, to my knowledge, reported for the first time - yet, to which extent they represent a relevant experimental model is questionable.
The paper is well written, and the results clearly presented. Yet the statistical methods are problematic, as with the small sample size parametric tests are likely inappropriate and although in methods a test for multiple comparisons is mentioned those results are not reported and in some experiments likely were not performed. In the revised manuscript these should be corrected.
References on rodent models of LPS-induced lung injury and NAC treatment are missing (like PMID: 18694812, 31631367, 16445696, and others)
Reviewer 2 Report
This study conducted by Drs. Chen and colleagues explored the potential therapeutic option for bovine respiratory disease (BRD) using LPS-induced acute lung injury model. The general idea is an interesting, however, reviewer would have to say the immature of the text submitted (e.g.: there are double Figure 4s and 5s displayed, and many typos appeared in text [L.61, 72, 122, 141, 215, 342, 351]). The description of “treatment” and “pre-treatment” were used confusingly in double Figure 4s and 5s, so please carefully use the appropriate one.
Seriously, although the studies were well performed, reviewer could not find clear association between in vitro and in vivo analyses (Figure 2 and second Figure 5 [should be corrected as Figure 7]) in this study.
Authors believe that the modulation of anti-inflammatory/oxidative properties in the lungs could ameliorate the LPS-induced ALI model, however, authors could not reproduce the changes in oxidative stress in EBTr cells with in vivo murine model.
More precisely, authors should clarify several uncertainties as follows:
1) LPS-induced HO-1 upregulation in EBTr cells (Figure 2C/D) was not reproduced in LPS-ALI model (Figure 7B), 2) LPS-induced GCLC downregulation in LPS-ALI model (Figure 7B) was not reproduced in EBTr cells (Figure 2C/D),
3) The effect of NAC pretreatment on significantly preserved Nrf2 expression after LPS exposure in EBTr cells (Figure 2D) was not reproduced (no asterisk) in LPS-ALI model (Figure 7C), and
4) Factors associated with oxidative stress such as CAT, Nrf2, SOD-1, GCLC did not differ between LPS and LPS+NAC groups in LPS-ALI model (Figure 7).
Although authors shows that the anti-inflammatory effect of NAC pretreatment in vitro and in vivo, anti-oxidative properties should be re-examined at multiple time points.
In these regards, reviewer did not think that the current model would be suitable for exploring the therapeutic options for BRD
Minor concerns:
1) line 160: Please cite relevant references or describe how much volumes of PBS wash.
2) line 170: In mice the left lung has only one lobe (no caudal lobe).
3) line 181: The details of all kits including MPO, MDA, CAT and SOD should be mentioned (Product company and Cat. number).
Reviewer 3 Report
The article aims to evaluate anti-inflammatory and antioxidant effects of N-acetylcysteine (NAC) pretreatment in LPS-induced models performed on bovine embryo tracheal cells and in female mice.
Authors demonstrated that NAC pretreatment effectively attenuated inflammation and oxidative stress what was confirmed by the use of a wide range of histopathological and analytical methods.
The manuscript is clearly and concisely written, however, I have some minor comments to the authors:
Section 2.3: Rephrase the paragraph and explain more in detail the methods of the establishment of the in vivo model. I suggest to add the Figure with graphical scheme with time relations of administration of pretreatments and LPS.
Section 2.3: Explain why only female mice were used in this study. Explain why NAC was given intragastrically and dexamethasone intraperitoneally.
Add more works of other authors on the use of NAC pretreatment or NAC treatment in LPS-induced models of acute lung injury into Introduction and Discussion sections – related to effects of NAC on inflammation and oxidative stress.
Explain all abbreviations in their first appearance in the text („ROS“ in line 65, „ICR“ in line 110, „GCLC, NQO1, HO-1, Nrf“ in lines 144-145, „AOEs“ in line 412, etc.).
Check again the English. There are some mistakes in the text which should be corrected („BRDC“ in line 41, „Thermo scientifice“ and „Dhoral hydrate“ in line 122, missing commas in line 386, etc.). In addition, some sentences should be rephrased, e.g. lines 72-76, lines 283-286, line 413-414.
Round 2
Reviewer 2 Report
Thank you for the opportunity to review the article exploring the effect of NAC on bovine respiratory disease. Authors conducted in vitro and in vivo studies based on their previous studies (Anim Biotechnol 2021, doi.org/10.1080/10495398.2021.1919129). In vitro studies using EBTr cells were well performed, and the results shown in Figures 2 and 3 were convincing, however, the associations of the results shown in the current manuscript between in vitro study using EBTr cells and in vivo murine models were not scientifically proven even in revised manuscript. Reviewer understand that this is not easy especially when using unestablished in vivo models, therefore optimization of in vivo study should be carefully performed based on the verification of multiple time points and/or different dosages of drug examined. Regretfully, the current manuscript did not include the updated data and did not scientifically demonstrate the several critical issues in reviewer's comments for previous manuscript as follows.
1) LPS-induced HO-1 upregulation in EBTr cells (Figure 3C/D) was not reproduced in LPS-ALI model (Figure 8B).
2) LPS-induced GCLC downregulation in LPS-ALI model (Figure 8E) was not reproduced in EBTr cells (Figure 3C/D).
3) The effect of NAC pretreatment on significantly preserved Nrf2 expression after LPS exposure in EBTr cells (Figure 3D) was not reproduced (no asterisk compared to LPS only) in LPS-ALI model (Figure 8C).
4) Factors associated with oxidative stress such as CAT, Nrf2, SOD-1, GCLC did not differ (no asterisk) between LPS and LPS+NAC groups in LPS-ALI model (Figure 8).
